



# Transport behavior displayed by water isotopes and potential implications for assessment of catchment properties

Dan Elhanati[1], Erwin Zehe[2], Ishai Dror[1], and Brian Berkowitz[1]

[1] Department of Earth and Planetary Sciences, Weizmann Institute of Science, Rehovot, Israel

[2] Institute of Water Resources and River Basin Management, Karlsruhe Institute of Technology (KIT), Karlsruhe, Germany

**Correspondence:** Brian Berkowitz (brian.berkowitz@weizmann.ac.il)

https://doi.org/10.5194/egusphere-2025-3365



**Abstract.**

Measurements of water isotopes are used routinely to estimate water transit time distributions and aquifer storage thickness in catchments. Water isotopes (e.g., $D_2O/H_2^{18}O$) are generally considered to behave identically to water molecules ($H_2O$); they are thus often considered fully representative of water movement and preferred over inert chemical tracers for catchment assessment purposes. However, laboratory-scale measurements presented here show that water isotopes exhibit transport behavior that is essentially identical to that of inert chemical tracers. The resulting measurements are then interpreted quantitatively, focusing on a comparative assessment of apparent mean water and mean tracer velocities, and the applicability of Fickian and non-Fickian (anomalous) transport models. For both water isotopes and inert chemical tracers, the measured mean tracer velocity is not necessarily equal to the apparent mean water velocity. It is thus critical to recognize this inequality when estimating catchment properties. For example, accounting for anomalous transport of water isotopes can significantly reduce overall estimates of aquifer storage thickness over an entire watershed.



## 1 Introduction

River catchments and streamflow play a pivotal role in water resources management (Sivapalan, 2018). A catchment represents a three-dimensional hydrological unit delineated by a watershed boundary, where precipitation is partly stored in the subsurface and partly released as evapotranspiration or runoff components, ultimately feeding streamflow. While the catchment water balance controls generation of streamflow amounts, catchments can be also regarded in analogy to chemical reactors (Grathwohl et al., 2013). Streamflow chemistry and contaminant fate are thus essentially controlled by the interplay of transport velocities and, in the case of reactive species, reaction rates of chemicals traveling through the catchment (Grathwohl et al., 2013; Berkowitz et al., 2016; Sternagel et al., 2021).

Natural catchments comprise inherent structural complexities above and below the land surface, which lead to heterogeneous spatial and temporal distributions of flow velocities. Accurately describing travel times in catchments is thus by no means straightforward (McDonnell et al., 2010). Travel (or transit) time distribution (TTDs) of *water*, defined as the durations water molecules require to traverse the catchment from rainfall to stream, are often regarded as a key metric for inferring streamflow chemistry (e.g., McGlynn et al., 2003; Weiler et al., 2003; Hrachowitz et al., 2013; Rodriguez et al., 2021; Benettin et al., 2022). A water TTD, and in particular the mean of the water TTD (from which one can infer the mean water velocity), is often used, for example, to estimate water storage and aquifer thickness in a catchment. However, it is difficult to uniquely define or determine a water TTD: clearly, one cannot directly measure the velocity of water molecules in an advective field.

More broadly, TTDs, sensu lato – e.g., TTDs of water, chemicals, and momentum – may represent different transport processes, which differ strongly with respect to the underlying mechanism and can also be time-dependent and substance-specific (Rinaldo et al., 2011). A common approach for inferring *water* TTDs of a catchment involves applying measurements of a tracer pulse transported by the water as input for a transport model. The normalized breakthrough curve of a unit mass input of the tracer thus corresponds to the tracer TTD, reflecting the distribution of fluid velocities and subscale diffusive mixing of tracer molecules between the flow lines (Simmons, 1982; Jury and Sposito, 1986). In this



context, models that describe various catchment transport processes are used to estimate
water TTDs from tracer breakthrough curves (e.g., McGuire and McDonnell, 2006; Bowers
et al., 2020; Sternagel et al., 2022; Wienhoefer et al., 2009; McDonnell et al., 2010;
Lischeid et al., 2000).
While measurements of any inert chemical tracer transported by the flow of water in a
catchment are often assumed to be suitable for inferring water TTDs, many studies focus
on use of ratios of isotopic tracers of the water molecule itself (i.e., the isotopologues
$H_2^{18}O$, $^2H_2O$, $^3H_2O$), because these molecules are considered to behave identically to $H_2O$
and they often enter the catchment naturally through rainfall (e.g., McDonnell and Beven,
2014; Rodriguez et al., 2021; Sternagel et al., 2022; Weiler et al., 2003; Aquilina et al.,
2006; Koeniger et al., 2010). They are therefore regarded as an optimal tracer of water,
compared to other chemical tracers (McGuire and McDonnell, 2006).
The above brief survey highlights the wide range of interpretations and methods related
to TTD assessment, particularly to estimates of *water* TTDs. Motivated by the literature
discussed above, the study here focuses on a frequently invoked, key assumption, namely
that isotopic tracers of the water molecule itself behave identically to $H_2O$ (with only
slightly different diffusion coefficients because of slightly different molecular weights) and
can therefore be used to infer the true mean water velocity and residence time in a porous
domain. For this purpose, water isotope tracer transport in a critical subset of a full
catchment – namely, the fully water-saturated domain – is examined. Breakthrough curves
of a water isotope and an inert chemical tracer are measured in macroscopically 1D porous
medium columns; the resulting curves are compared and subsequently interpreted
quantitatively. A fundamental question is thus studied: What are the implications of using
water isotopes as tracers, as compared to inert chemical tracers, in terms of defining a water
TTD and its mean?

**2 Methods**
A laboratory-scale experimental setup was constructed to compare the transport
behavior of inert chemicals to the transport behavior of water isotopes. This setup aimed
to examine flow and transport in a controlled saturated porous medium, allowing the
measurement and comparison of the tracer (Br) and a water isotope tracer (water containing



a higher $D_2O/H_2O$ ratio than commonly found in nature) in a flow regime which exhibits
anomalous transport.

In a previous study, Elhanati et al. (2023) examined the transport behavior of an inert

chemical tracer (Gd) in porous medium columns, under time-dependent velocity conditions
in a macroscopically 1D flow regime. The same experimental setup was adapted for this
study as it consistently showed anomalous transport for different flow rates and porous
medium arrangements. The setup consisted of three vertical columns measuring 100 cm in
length with an internal radius of 1.4 cm, packed with a fully water-saturated porous medium
composed of clean quartz sand grains, and with water and tracer injected from below. First,
three effectively (macroscopically) homogeneous porous medium columns (Columns I-III)
were packed uniformly with sand having an average grain size of 1.105 mm (mesh size
12/20), and porosity of 0.38. Subsequently, the three columns were cleaned, and each was
packed with an alternating pattern of three different sand sizes (Columns IV-VI) to produce
a heterogeneous porous medium (see Elhanati et al., 2023 for a full description of the
experimental setup). Elhanati et al. (2023) reported that both packing configurations (i.e.,
homogeneous and heterogeneous) displayed long tailing in the Gd breakthrough curves
and other behavior indicative of anomalous transport.

Three solutes were used for this study: (1) NaBr salt, an inert tracer used as a benchmark,

with an initial concentration of 10 ppm Br; (2) $D_2O$, with an initial concentration of 10,000
ppm D; (3) A combination of both solutes (NaBr and $D_2O$). Repeating the experiments
with a combined solute of $D_2O$ enriched water and Br allowed comparison of the resulting
breakthrough curves. Although an interaction of the two tracers was not expected, the
experiment was repeated with only the $D_2O$ for validation. Each experiment was conducted
simultaneously in the three columns, starting with a short solute injection before switching
to double-distilled water flow for the rest of the experiment. Samples were collected at the
column outlets using a fraction collector.

First, experiments were conducted in a homogeneous medium using coarse-grained sand

for the three solutes mentioned above (experiment sets A1: NaBr+$D_2O$, A2: $D_2O$, and A3:
NaBr, respectively; $Q = 1.0$ mL/min). Next, the homogeneous experiments were repeated
with the combined NaBr and $D_2O$ solute with a higher flow velocity, namely double
volumetric discharge (experiment set A1$_{fast}$: NaBr+$D_2O$; $Q = 2.0$ mL/min), to test for





consistency in different flow conditions. Subsequently, two sets of experiments in a
heterogeneous medium for the slow (experiment set B1: NaBr+$D_2O$) and fast (experiment
set B1$_{fast}$: NaBr+$D_2O$) flow conditions were performed. See Table 1 for a summary of the
experiments. To add perspective, a representative Peclet number (Pe) for the experimental
set-up can be estimated. Here, Pe = $L\bar{v}_w/D$, where $L$ is a characteristic length, chosen here
as the average grain size diameter ($L$ = 0.11 cm), $\bar{v}_w$ is the average local flow velocity,
and $D$ is a mass diffusion coefficient ($D$ = $2\times10^{-5}$ cm$^2$/s, representative of typical inert
anionic tracers like bromide). For the uniformly packed sand columns with $Q$ = 1.0
mL/min, $\bar{v}_w$ = $7.1\times10^{-3}$ cm/s (see calculation in Sect. 3.2), so that Pe ≈ 39.

**Table 1.** Set-up and conditions of column experiments.

| Experiment | Packing | Flow Rate | Solute |
|---|---|---|---|
| A1 | Uniformly packed sand (Columns I, II, III) | 1.0 mL/min | NaBr+$D_2O$ |
| A2 | uniformly packed sand (Columns I, II, III) | 1.0 mL/min | $D_2O$ |
| A3 | Uniformly packed sand (Columns I, II, III) | 1.0 mL/min | NaBr |
| A1$_{fast}$ | Uniformly packed sand (Columns I, II, III) | 2.0 mL/min | NaBr+$D_2O$ |
| B1 | Alternating pattern of three different sand sizes (Columns IV, V, VI) | 1.0 mL/min | NaBr+$D_2O$ |
| B1$_{fast}$ | Alternating pattern of three different sand sizes (Columns IV, V, VI) | 2.0 mL/min | NaBr+$D_2O$ |


Water samples were measured by Inductively Coupled Plasma-Mass Spectrometry
(ICP-MS; Agilent). The ICP-MS ionizes the samples and detects the presence of specific
atomic masses, which allows the determination of the concentration of the Br and D at the
column outlet throughout the experiment. While isotopes of light elements are not readily
measurable by ICP-MS due to low ionization efficiency and spectral interference, it is
possible to measure deuterium-containing polyatomic species (e.g., ArD$^+$) as an accurate





proxy for D analysis (Galbács et al., 2020). In excess of $H_2O$, $D_2O$ rapidly converts to HDO
in equilibrium ($D_2O + H_2O \rightleftarrows 2HDO$), which is linearly correlated to the measurable $ArD^+$
ion in the plasma. While the measured Br background concentration is below the
instrument detection limit, the double distilled water applied in the experiment comprises
a naturally occurring $D_2O/H_2O$ ratio. The background concentration was subtracted from
the results presented to show the breakthrough of the Br and $D_2O$ solutes over their
naturally occurring background concentration.

**3 Results and discussion**
**3.1 Comparison of Br and $D_2O$ breakthrough curves**
The slow-flow homogeneous porous medium experiments show similar behavior for the
enriched-deuterium water and the bromide tracer (Fig. 1a). Both signals show a similar
breakthrough curve, as demonstrated by the onset and length of the breakthrough measured
at the outlet. This finding is consistent for the bromide and $D_2O$ when injected as a single
chemical species or when combined in a single experiment. This coupling establishes the
similar behavior of the two chemical species, independent of a dynamic resulting from the
simultaneous injection. This finding is apparent for all three columns, which show the same
behavior in different packing arrangements. Column III, in particular, displayed longer tails
of the breakthrough curves, consistently for both the Br and the $D_2O$. This can be seen, in
particular, in the semi-log scale which allows a focus on the long-tailed behavior. The faster
flow experiments also showed consistency of this finding across the three columns (Fig.
1b). The heterogeneous porous medium experiments show longer tails compared to the
homogeneous porous medium experiments, for both the slow flow and high flow
experiments. For both flow scenarios, the bromide tracer and $D_2O$ water displayed similar
breakthrough curves for each column (Figs. 1c and 1d). The various replicate experiments
shown in Fig. 1 illustrate natural variability, which is exhibited particularly in the behavior
of the long-time tails for each specific column and flow rate.
The results of both the heterogeneous and homogeneous column experiments, and for
different flow rates, show that water isotopes behave similarly to inert chemical tracers.
This finding is discussed in detail in Sect. 3.2 and Sect. 3.3. In both the homogeneous and





heterogeneous columns, a similar discharge was experimentally maintained, resulting in
similar mean water travel times. However, it can be seen that due to the differences in
medium composition, the heterogeneous columns displayed longer tails in the measured
breakthrough. The longer tails seen in the heterogeneous medium compared to the
homogeneous medium, are found for both the Br tracer and the deuterium-enriched water,
indicating that both solutes indeed represent a transport process.

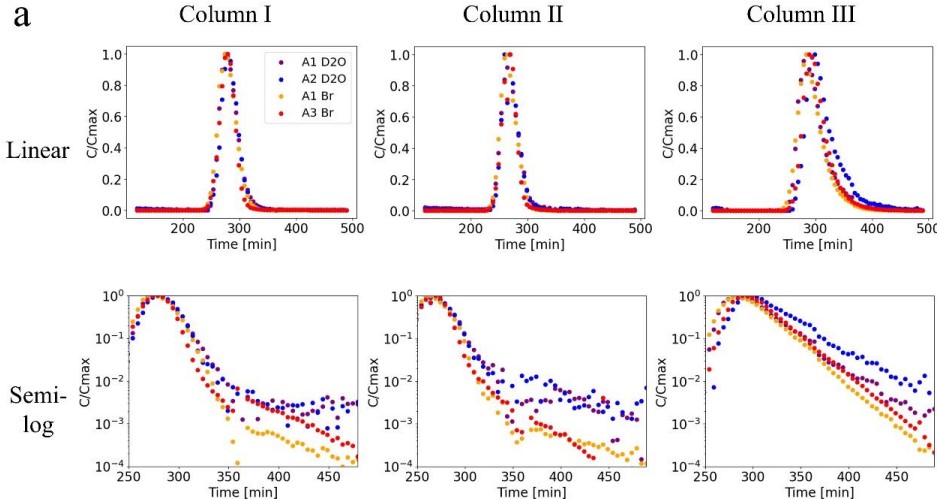




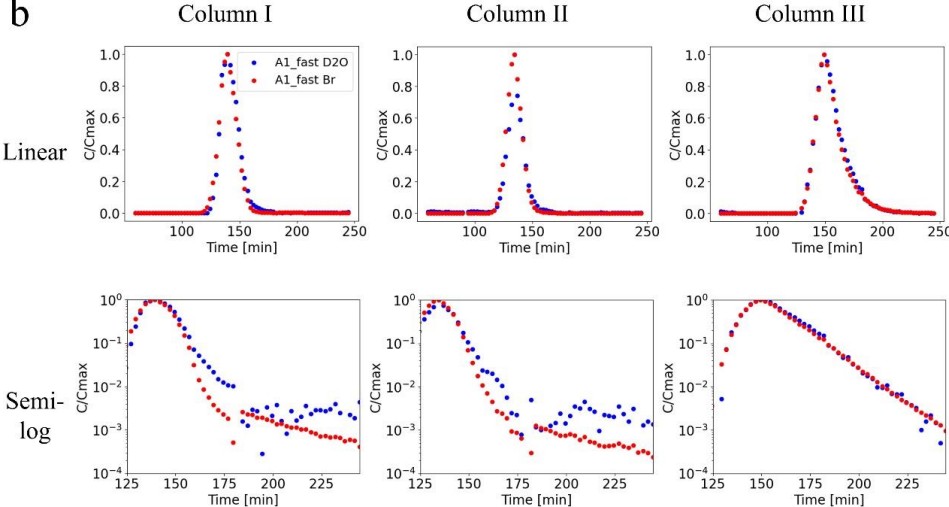


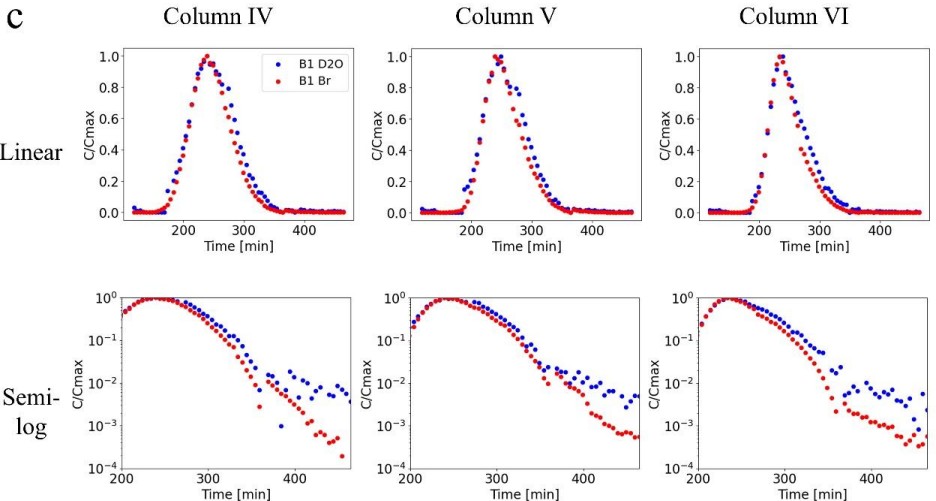




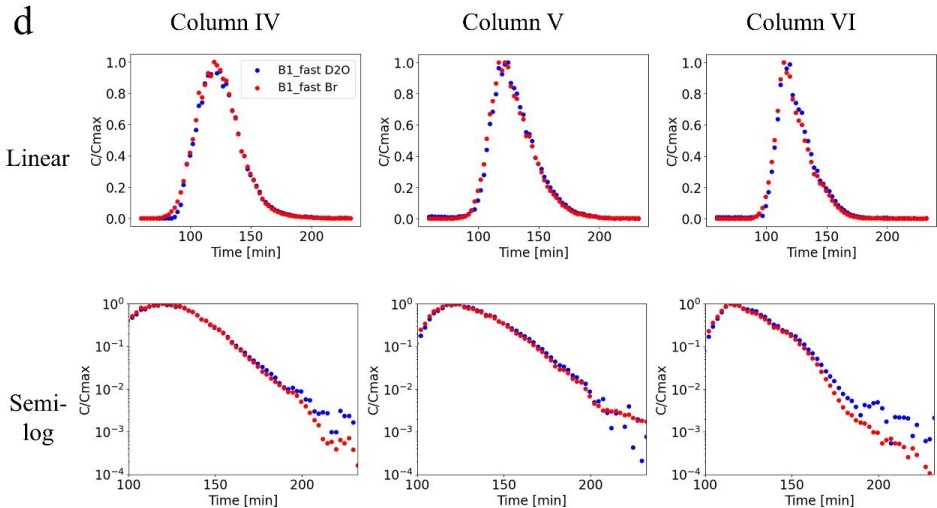

**Figure 1.** Breakthrough curves (A1, B1: $D_2O$+Br; A2: $D_2O$; A3: Br) for the three homogeneous and the three heterogeneous porous medium columns in linear scale (top row) and semi-log scale (bottom row). **(a)** homogeneous slow-flow experiments (A1: NaBr+$D_2O$, A2: $D_2O$, A3: NaBr); **(b)** homogeneous fast-flow experiments (A1$_{fast}$: NaBr+$D_2O$); **(c)** heterogeneous slow-flow experiments (B1: NaBr+$D_2O$); **(d)** heterogeneous fast-flow experiments (B1$_{fast}$: NaBr+$D_2O$). $D_2O$ concentrations at the tailing end of the breakthrough are generally slightly higher than Br concentrations, because the naturally occurring $D_2O$/$H_2O$ ratio fluctuates around the measured background value. Note the different time scales between experiments.

## 3.2 Water and tracer transport in a fully water-saturated porous media: a Gedanken experiment

While one cannot directly measure velocity of water molecules, an *apparent average water velocity*, $\bar{v}_w$, which represents an average macroscopic value over the entire medium, is commonly determined by use of Darcy's law. In a macroscopically 1D column, for example, $\bar{v}_w$ can be determined by the simple relation $\bar{v}_w = Q/nA$, where $Q$ is fixed volumetric discharge, $n$ is an effective porosity (e.g., determined by comparing weights of a sand or rock core sample under dry and then water-saturated conditions), and $A$ is the cross-sectional area of flow.

A critical question then arises: is the average velocity of an inert chemical tracer, $\bar{v}_T$, identical to that of the apparent average water velocity, $\bar{v}_w$? In principle, the answer is, in





general, no (i.e., $\bar{v}_w \neq \bar{v}_T$), unless the domain is perfectly homogeneous over the length
and/or time scales of measurement (e.g., Cortis et al., 2004).

A Gedanken experiment is useful to visualize this important difference. In a fully water-

saturated porous column containing pore-scale heterogeneities, or very small-scale, lower
permeability inclusions embedded in the column, the flow and transport are
macroscopically 1D (Fig. 2). The behavior of the water and the migration of an inert tracer,
such as Br⁻, can be determined theoretically. By estimating the porosity and cross-sectional
area of flow through the column, and for a fixed $Q$, $\bar{v}_w$ can be calculated using Darcy's law,
and the mean travel time of water through the column can be estimated by dividing the
column length by $\bar{v}_w$. Measuring the Br⁻ ions, however, will expectedly yield a different
result: if a pulse of $X$ Br⁻ ions is injected into the column, $Y$ Br⁻ ions may reach the
inclusions and remain there for a very long time. Estimation of the mean velocity, $\bar{v}_T$, of
the Br⁻ ions at the column outlet would thus result in lower velocity compared to the
apparent mean water velocity, due to the slow-moving ions (i.e., a long-tail in the tracer
breakthrough measurement).

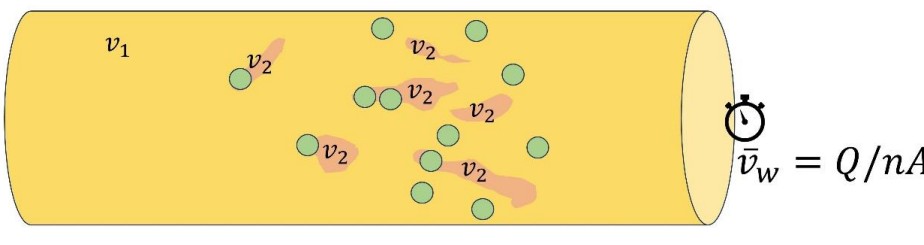

Flow direction


**Figure 2.** Conceptual macroscopically 1D flow through a 1D porous medium column with
a mean velocity ($\bar{v}_1$) containing (very small-scale or pore-scale) lower velocity inclusions
($\bar{v}_2$). The Br⁻ ions injected as a pulse into the column are marked with green circles.
Calculation of the apparent mean water velocity ($\bar{v}_w$) at the column outlet will yield a
higher velocity than the actual velocity of the transported Br⁻ ions due to the lower
velocities experienced by some Br⁻ ions in the inclusions.

This Gedanken experiment leads to the expected conclusion that the mean velocity of

the chemical tracer does not necessarily represent the apparent mean water velocity, due to





even small-scale mobile and immobile zones in the medium. However, it is also clear that
the origin of the measured behavior of the inert tracer is the direct result of the measurement
process. One can replace the Br⁻ ions in the above example with water isotopes and reach
the same result: some of the tagged water molecules will reach the inclusions and the
estimated mean tracer velocity $\bar{v}_T$ will be slower than the apparent mean water velocity
$\bar{v}_w$. The act of tagging water effectively changes a water molecule to act as a "non-water"
tracer, in the context of breakthrough measurements; in other words, the measured velocity
represents the mean velocity of a tracer (be it a water isotope or an inert chemical), rather
than the apparent mean velocity of the water.

### 3.3 Quantification of transport behavior


It is important to recognize the inherent difference between $\bar{v}_w$ and $\bar{v}_T$ when quantifying
tracer transport in a fully water-saturated porous medium, In a macroscopically one-
dimensional domain, the apparent mean water velocity $\bar{v}_w$ does not represent the actual
travel times of all water molecules through the medium, but rather an average macroscopic
value over the entire medium. On the other hand, an inert tracer transported by the water is
subjected to advection and hydrodynamic dispersion, as well as to subscale diffusive
mixing. As the chemical tracer is transported through the medium, it displays a distribution
of velocities which represents a fingerprint of the heterogeneous flow paths. Therefore, the
transport of a tracer inherently reflects a distribution of velocities for which $\bar{v}_T$ represents
the mean.
With this understanding, how does one interpret and quantify experimental results such
as those discussed in Sect. 3.1? In an effectively (macroscopically, continuum-level)
homogeneous porous medium, the tracer particles can in essence display Fickian dispersion
and $\bar{v}_w = \bar{v}_T$ (Berkowitz et al., 2006). In this situation, the classical 1D form of the
advection-dispersion equation (ADE) for steady-state flow can be applied to quantify the
transport dynamics, $\partial C/\partial t = -v\, \partial C/\partial x + D^* \partial^2 C/\partial x^2$, where the by velocity $v$ is by
definition based on $\bar{v}_w$ and $D^*$ is a dispersion coefficient. However, in many cases, the
velocity distribution often gives rise to non-Fickian (or anomalous) transport, which can
be manifested by, e.g., the occurrence of long tails in measured breakthrough curves (e.g.,

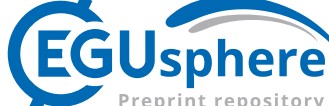

259 Cortis et al., 2004; Berkowitz et al., 2006). Thus, the effect of anomalous transport may be

260 significant for mean TTD estimation, which might differ substantially from the apparent

261 mean water velocity; this is discussed in Sect. 3.4.

262  Here, the continuous time random walk framework (CTRW) was used to interpret

263 measured breakthrough curves (Berkowitz et al., 2006) such as shown in Fig. 1. The CTRW

264 represents a continuum-scale, ensemble average behavior relevant to the interpretation of

265 these macroscopically 1D column experiments; it is especially suitable for this task because

266 it inherently employs $\bar{v}_T$ in its formulation. Solutions based on CTRW have been shown to

267 yield a good description of non-Fickian transport scenarios (Dentz et al., 2008, 2018, 2023;

268 Edery et al., 2015; Bijeljic et al., 2011, 2013; Nissan and Berkowitz, 2019; Goeppert et al.,

269 2020).

270  In a CTRW particle tracking (PT) formulation, applied here, probability density

271 functions stochastically define particle transitions in space and time (see Elhanati et al.,

272 2023 and Nissan et al., 2017 for a complete mathematical description). A truncated power

273 law distribution is assigned for the temporal probability density function, defined with the

274 exponent $\beta$, which is a measure of the non-Fickian nature of the transport (Nissan et al.,

275 2017). A power law exponent of $\beta > 2$ implies Fickian, or essentially Fickian, behavior for

276 which an ADE solution is generally applicable (Berkowitz et al., 2006); $\beta < 2$ is a descriptor

277 of non-Fickian transport. The first spatial moment of the chemical species plume in the

278 flow direction, $v_\psi$, is defined as the mean particle velocity and is therefore applied herein

279 as the mean tracer velocity, $\bar{v}_T$. It is noted that the breakthrough curves from the

280 experiments presented here were expected to display anomalous transport based on

281 experiments and analysis reported previously by Elhanati et al. (2023). Anomalous

282 transport even in macroscopically homogeneous porous media arises because of subtle,

283 residual pore-scale disorder effects, with diffusion into pore-scale stagnant regions that can

284 lead to a wide (power law) distribution of travel times (Cortis et al., 2004; Berkowitz et al.,

285 2006).

286  Two key characters are assessed in this context, focusing on a representative

287 breakthrough curve: the mean velocity of the $D_2O$ tracer and the nature of the long tails in

288 the breakthrough curves. As an example, the slow flow homogeneous porous medium $D_2O$

289 (A1) dataset was interpreted using the CTRW-PT formulation discussed above. An





approximate fit to the data yielded a power law exponent of $\beta \approx 1.84$ (Fig. 3), which is
indicative of anomalous transport.
Moreover, the CTRW-PT simulation yielded a mean tracer velocity of $\bar{v}_T = 5.8 \times 10^{-3}$
cm/s. In sharp contrast, for this experiment, the apparent mean water velocity can be
directly determined as $\bar{v}_w = 7.1 \times 10^{-3}$ cm/s for the experiment parameters ($Q = 1.0$ mL/min,
$n = 0.38$, and $A = 6.16$ cm$^2$). Clearly, $\bar{v}_w \neq \bar{v}_T$. Figure 3 also shows two representative fits
of the data using a 1D solution of the classical ADE. Here, using the value of $\bar{v}_w = 7.1 \times$
$10^{-3}$ cm/s, which according to the theory underlying the Fickian-based ADE is the relevant
velocity, the solution is seen to strongly over-estimate the advance of the peak, and to be
unable to capture the long-time tailing. Furthermore, even if the mean velocity in the ADE
is – incorrectly in this case – chosen to match the peak travel time of the data (simple
inspection of the breakthrough curve indicates a peak travel time of about 274 min), the
ADE solution is unable to capture the early arrivals and the long-time tailing (Fig. 3). In
this case, the mean velocity is approximately $6.0 \times 10^{-3}$ cm/s, and the dispersion coefficient
was chosen to yield a breakthrough width similar to that using $\bar{v}_w$.
Finally, as an additional consideration, note that with a column length of 100 cm, the
peak travel time of 274 min can be translated to an overall assumed "mean" velocity of
about $6.1 \times 10^{-3}$ cm/s, which is close to the estimate of $\bar{v}_T = 5.8 \times 10^{-3}$ cm/s from the
CTRW-PT simulation, and clearly distinct from the apparent mean water velocity $\bar{v}_w = 7.1$
$\times 10^{-3}$ cm/s. Moreover, from the same breakthrough curve, the velocity corresponding to
the time required for 50% of the tracer to be eluted from the column is essentially identical
to $\bar{v}_T$.





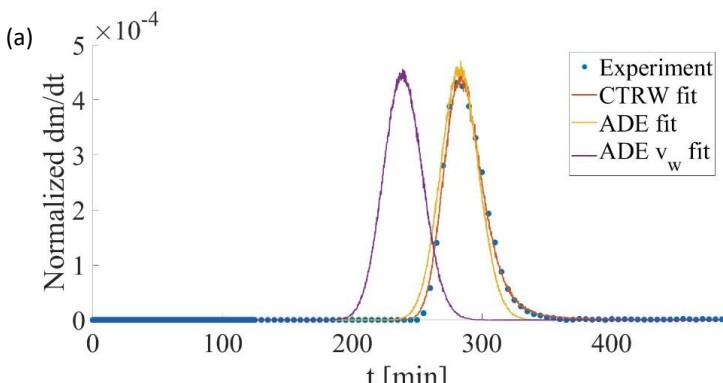



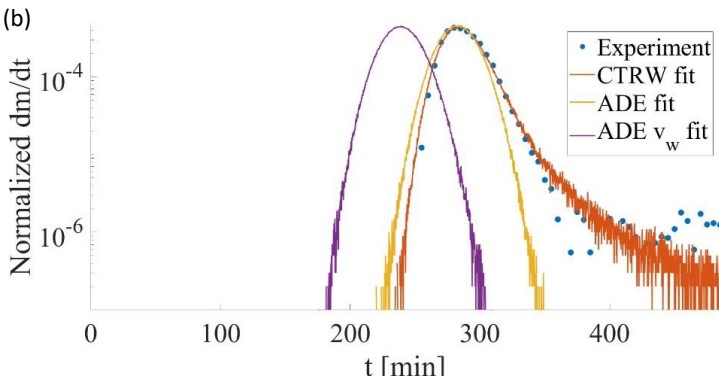


**Figure 3.** Comparison of experiment and CTRW simulation breakthrough curves for the
D$_2$O slow flow homogeneous porous medium column (linear scale **(a)**). The long tails
indicative of anomalous transport can be seen in the semi-log scale **(b)**, and in the modeled
power law exponent ($\beta$=1.84). CTRW solution: $\bar{v}_T = 5.8 \times 10^{-3}$ cm/s (with a generalized
dispersion coefficient of $0.14 \times 10^{-3}$ cm$^2$/s). Two solutions of the ADE are also shown, one
using the value of $\bar{v}_w = 7.1 \times 10^{-3}$ cm/s ("ADE v$_w$ fit"; with a dispersion coefficient of $D^*$
$= 1.4 \times 10^{-3}$ cm$^2$/s), and a fit to the peak of the data, yielding a velocity of $6.0 \times 10^{-3}$ cm/s
("ADE fit"; with $D^* = 0.8 \times 10^{-3}$ cm$^2$/s).

## 3.4 Implications of using isotopic waters for inferring catchment properties

The above analysis – for a macroscopically 1-D column – can be used to provide a first

assessment of the impact of using differing estimates of a mean velocity on catchment





travel time estimates. It should be recognized at the outset that while catchments are inherently complex, highly heterogeneous 3D systems, involving surface water, soil layer, and aquifer components, catchment assessments are often based on largely 1D conceptualizations, accounting for a *bulk* water and tracer input (recharge) region and an ultimate discharge measured at a conveniently monitored outlet point (spring or stream) (e.g., Stewart et al., 2010; Koeniger et al., 2010; Benettin et al., 2022; Rodriguez et al., 2021).

In the column experiments discussed in Sect. 3.1, it was shown that water isotopes migrate like inert tracers. Furthermore, it was shown in Sect. 3.3 that the tracer transport displays non-Fickian behavior, with $\bar{v}_w \neq \bar{v}_T$ and longer-than-Fickian breakthrough tailing, both of which affect the assessment of travel and residence times. In catchment assessment, it is assumed that an often ambiguously defined mean travel time of water exists and that it can be represented by considering a hydraulic retention time, which is defined as a storage volume divided by a volumetric flow rate. Here, it is seen that the hydraulic retention times are distinct from average residence times: one cannot observe the migration of individual water molecules, but only the migration of tracers – whether inert chemicals or water isotopes – in water. This implies that trapping of water isotopes in low conductive regions will induce strong differences between estimated hydraulic retention times and average isotope travel times.

Returning to the specific example calculation discussed in Sect. 3.3, mean travel times over a 1 cm length are therefore ~140 s and ~172 s, for the apparent mean water velocity ($\bar{v}_w = 7.1 \times 10^{-3}$ cm/s) and the mean tracer velocity ($\bar{v}_T = 5.8 \times 10^{-3}$ cm/s), respectively. The mean particle transit time is longer than the water travel time due to the dispersion-induced anomalous transport by a ratio of ~1.2, for these specific experiments.

Returning now to the background discussed in the Introduction, a key parameter of interest in catchments is the assessment of the aquifer (fully water-saturated region) storage thickness over an entire watershed; see, e.g., Stewart et al. (2010) for extensive discussion of aquifer storage considerations. In this context, Stewart et al. (2010) provide an extensive summary of published studies that report mean water isotope travel times in macroscale catchments. The authors note that the various catchments appear sufficiently large to yield





relatively similar average behaviors. Moreover, the assessments all suggest the existence
of substantial storage volumes for recharge water into the aquifer zone.
These mean travel times estimates, which actually represent tracer transport and thus
$\bar{v}_T$, are all based on analysis of $^3$H and assumed in these publications to represent the
apparent water velocity, $\bar{v}_w$. [It is recognized, parenthetically, that different isotopes are
likely to yield somewhat different average travel times, as may different inert chemical
tracers with different masses and rates of diffusion, but this factor is not relevant for the
key points and the short-term column experiments reported here.] If the tracer transport
were Fickian, then this estimate would indeed represent $\bar{v}_T$. However, real aquifers
generally display non-Fickian behavior (e.g., Goeppert et al., 2020; Dentz et al., 2023), and
in the example calculation given in Sect. 3.3, the true value of $\bar{v}_w$ is a factor of ~1.2 faster
than $\bar{v}_T$. Stewart et al. (2010) conclude their analysis with an example calculation of a
catchment aquifer storage thickness, based on their summary of many catchments. They
posit a catchment with annual precipitation of 1000 mm, evapotranspiration of 600 mm
(and thus annual recharge of 400 mm/year), and 50% aquifer flow in a formation with an
overall porosity of 20%. For an estimated (apparent) mean (water) travel time of 10 years,
a 10 m aquifer thickness is needed over the entire watershed to account for the long travel
times in the data. However, given that water isotopes do not directly represent the water
mean travel times and yield longer travel times, the actual aquifer thickness may be lower.
The ratio of 1.2 between apparent mean water and mean tracer travel times calculated above
applies only to the specific columns studied here. However, applying the calculated ratio
for the scenario presented, to give a coarse estimate, would yield a significantly smaller
aquifer thickness of ~8 m. While the ratio of water and tracer mean travel times should be
estimated for any given scenario separately, the example above demonstrates the
importance of this estimation in inferring aquifer characteristics.

**4 Conclusions**
The experimental findings demonstrate the similarity between the measured transport
behavior of water isotopes and an inert chemical tracer in fully water-saturated porous
media. This similarity is evident across different flow velocities and porous medium



compositions. Notably, water isotopes exhibit the same transport behavior as tracers; the
very act of *tagging* water molecules, implicit in the measurement of any water isotope,
yields a measurement of their migration as a chemical tracer, which is not identical to the
bulk water flow. Moreover, the experiments here demonstrate that even in relatively
homogeneous sand columns, both water isotopes and inert chemical tracers exhibit non-
Fickian (anomalous) transport, and the mean tracer velocity is not necessarily equal to the
apparent mean water velocity. Consequently, studies that rely on water isotopes to estimate
catchment properties like water TTDs and aquifer storage thickness must recognize this
subtle but critical inequality between apparent mean water and mean tracer velocities, and
not use them interchangeably to represent the actual travel times of tracers and water
isotopes.

*Data availability.* The data on which this article is based are available online on Zenodo:
https://zenodo.org/doi/10.5281/zenodo.12187848 (Elhanati et al., 2024).

*Competing interests.* At least one of the (co-)authors is a member of the editorial board of
Hydrology and Earth System Sciences. The authors have no other competing
interests to declare.

*Author contributions.* DE, EZ, ID, and BB formulated the ideas that resulted in the project,
defined the goals and aims of the study, and contributed to the various study components.
DE and BB developed the experimental methodology, ID and DE developed the isotope
analytical measurement protocol, and DE implemented the methodology and carried out
the data analysis. DE and BB drafted the initial manuscript. All authors took part in
reviewing and editing the final manuscript.

*Acknowledgments.* B.B. thanks the Minerva Foundation for support. D.E. gratefully
acknowledges the support of the Weizmann Institute for Environmental Sustainability.
B.B. holds the Sam Zuckerberg Professorial Chair in Hydrology.

 

*Financial support.* B.B. and E.Z. gratefully acknowledge support through the ViTamins
project, funded by the Volkswagen Foundation (Grant No. AZ 9B192).

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
