# Peer review of "Transport behavior displayed by water isotopes and"

_EGUsphere, 2025_

## Author Comment (AC1)

**Response to referee #1:**

Thank you for giving me the opportunity to read your manuscript. It is well written and interesting. Your results provide novel and important information to the scientific community. You did a great job describing the rather complex methods in a way that is easily understood by the readers. Your discussion on the implication of the findings to catchment water movement is well described, and provides direct, actionable, and targeted advice to hydrologic planners. Overall, I was very impressed and I only have minor revision requests:

**RESPONSE:** We thank the referee for the highly positive appraisal of our manuscript; we are particularly pleased that the description was easily understood. We appreciate the suggestions for minor revisions, and are pleased to incorporate them in the revised manuscript. We detail our responses point-by-point, below.

Line 79 Introduction: I recommend breaking the last paragraph of the introduction into two paragraphs. The first should provide more literature on chemical tracers (namely Br and D2O/H2O, used in your study) and any other literature that has studied on water transport as it relates to chemical tracer behavior. Additionally, this should include the literature of Fickian and non-Fickian (anomalous) transport models and how they apply to water behavior, as you have mentioned in your abstract. The second paragraph should describe your objectives and include the more specific explanation of your methods shown in lines 79 – 85.

**RESPONSE:** We have revised the introduction to expand upon the topics suggested by the reviewer. Namely, we have added a paragraph to further discuss the use of inert chemical tracers and included four new references: "Measurements of any inert chemical tracer transported by the flow of water in a catchment are often assumed to be suitable for inferring water TTDs. Ion tracers, such as bromide, have been used widely to study chemical transport in natural catchments, as they are relatively inexpensive and easy to measure (Levy and Chambers, 1987; Bowman, 1984). Their use has since been expanded to investigate broader aspects of catchment hydrology, including groundwater recharge and evapotranspiration (Chen et al., 2021), and catchment baseflow (Rai and Iqbal, 2015)." In addition, text was expanded in the subsequent paragraph, to describe the nature of non-Fickian transport and its relation to conservative tracers vs. water isotopes: "Moreover, in many catchments subject to chemical transport with relatively high Peclet numbers, conservative chemical tracers exhibit long-tailed breakthrough curves, a non-Fickian behavior that cannot be explained by the traditional

advection-dispersion equation (ADE). This behavior, also referred to as anomalous transport, arises from the heterogeneous nature of the porous media through which the tracer plume travels (Berkowitz et al., 2006a). Water isotopes are therefore regarded as an optimal tracer of water, compared to other chemical tracers, as they are assumed to essentially represent water flow and not chemical transport (McGuire and McDonnell, 2006)." Finally, as suggested by the referee, the last paragraph of the Introduction was split into two, to better highlight the objectives and methods. The description of the methods was expanded to note the use of porous media columns that are known to exhibit non-Fickian transport conditions, and to mention consideration of related catchment properties.

Line 258: Explain Fickian and non-Fickian transport in 1-2 sentences.

**RESPONSE:** As suggested, we have added text to clarify the difference between Fickian and non-Fickian transport: "However, in many cases, the velocity distribution often gives rise to non-Fickian (or anomalous) transport, which can be manifested by, e.g., the occurrence of long tails in measured breakthrough curves (Cortis et al., 2004), which cannot be captured by the traditional implementation of Fick's law in the ADE that assumes a symmetrical temporal breakthrough curve." See also the previous comment, for text added to describe non-Fickian transport in the Introduction.

Conclusion: I recommend providing one more paragraph summarizing the findings that your results may indicate smaller aquifer thickness requirements. I also recommend providing a sentence or paragraph on the applicability of this research to global aquifers/catchments to provide a global perspective/conclusion to this research.

**RESPONSE:** As suggested, we have split the Conclusions into two paragraphs, expanding the second to focus on the implications for estimates of aquifer thickness in catchment studies and the possible implications at the global scale: "Consequently, studies that rely on water isotopes to estimate water TTDs must recognize this subtle but critical inequality between apparent mean water and mean tracer velocities, and not use them interchangeably to represent the actual travel times of tracers and water isotopes. Our findings also indicate that selecting the correct velocity for aquifer thickness calculations can yield significantly smaller thickness estimates— an effect that could have even greater implications when applied at the global scale, beyond the illustrative example shown here."

---

## Author Comment (AC2)

**Response to referee #2:**

The title of this paper "Transport behavior displayed by water isotopes and potential implications for assessment of catchment properties" is misleading, as it suggests the paper has something to say about water isotopes and their use in assessing catchment properties. I see no evidence of this whatsoever.

**RESPONSE:** The referee states that the title is misleading and makes a sweeping statement regarding the nature of our analysis and discussion regarding use of water isotopes for assessment of catchment properties. As we detail below, we respectfully disagree with this statement and provide explanations that justify the manuscript title and our analysis. The manuscript deals *explicitly* with transport behavior of water isotopes – presenting experiments and a quantitative discussion – and then makes *explicit* reference to catchment properties and a method of analysis, illustrating a quantitative implication regarding assessment of aquifer storage thickness. Nonetheless, we have introduced additional clarifications and explanations (as noted below) based on the referee's comments, where appropriate, in the revised manuscript.

The core claims of the paper rest on two points:

     1. stable water isotopes behave similarly to inert tracers like Br-

     2. The v_T parameter of a CTRW model fit to breakthrough curves showing non-fickian behavior is not the same as the value of v_W calculated from v_W=Q/(nA).

I have no major issue with either of these points, per se. However the authors then argue that there is some issue with using isotopes to understand water movement through catchments -- as though the movement of water molecules were somehow different from the movement of "water" itself.

**RESPONSE:** We are glad that the referee has no major issues with these two points, although regarding point 2, we explain below that the issue relates to a general question not limited to the CTRW analysis.

The referee's difficulty arises in the analysis and interpretations that follow these two points. Here the referee implicitly takes the position that there is no issue or difficulty in using water isotopes to understand water movement through catchments. Our line of argument requires elaboration on this point, which is given in the manuscript and expanded upon here. *The issue*

*here is that Darcy's law does not actually identify and measure the velocities of all water molecules (e.g., including those that may be trapped in an immobile zone for extremely long or essentially infinite times); rather, Darcy's law offers a means to determine an "effective", continuum-scale, mean velocity of "water". Recognition of this difference between the definitions of the "mean water velocity" (given by Darcy's law), and the mean velocity of water isotopes or chemical tracers is a core feature of our analysis.* Briefly, we can define "movement of water itself" by using Darcy's law (as described in the manuscript, Section 3.2, paragraph 1). This is, indeed, the mean movement of "water".

Significantly, though, the catchment literature (citations given in the manuscript) often suggests or states that use of water isotopes to estimate travel times and, in particular, a mean travel time, yields a mean travel time of *water*. However, we demonstrate in the experiments presented in the manuscript that it this is in fact not the general case. Isotopes yield mean travel times similar to those of inert chemical tracers, as we show in experiments reported in the manuscript (and accepted by the referee). We thus show that v_T, the mean travel time of the isotope and the tracer, is distinct from the mean travel time of the "water" itself, as determined from Darcy's law. And as we explain further in the manuscript, this key point is often not recognized, so that frequent application of the classical ADE, in particular, to interpret breakthrough curves can be misleading and in fact incorrect. We then show that for the particular experiments under consideration, the ADE model is not adequate while a more general CTRW model (which encompasses the ADE as a special case) can interpret the data (Figure 3). Thus, with regard to the second point highlighted above by the referee, the fact that v_T is distinct from v_W is *not* specific only to the CTRW model. Significantly, it is equally relevant to the advection-dispersion equation (ADE) model, and similar approaches. We expand further on this point in the next comment/response below.

**_DONE:_** In light of the referee comments and our responses above, we have added a statement in the revised manuscript to clarify the arguments already provided therein. In Section 3.2., first paragraph, we have added the text highlighted above in *italics*. Additional text has been added related to v_W and v_T, and ADE and CTRW models; see the next comment/response /"DONE" text below.

To me the issue seems to rest entirely on the assumption that the v_T parameter that arises in CTRW theory is the 'true' mean velocity of the tracer, and that this ought to correspond with the value of v_W calculated as mentioned. Tf there is a mismatch between theory and observation then the issue is not with reality, it is with the theory . If the discrepancy between

the two 'velocities' arises regardless of which tracer is used (which they themselves say their results support) then the use of isotopes as a tracer is not at issue. Rather it seems to me it highlights a theoretical confusion about what v_T and v_W actually mean in relation to one another. Does CTRW theory assert that they ought to be the same? Are the observations in agreement with the theory, or at odds with it? Where exactly does this difference come from? It certainly seems like a paper that disentangles that issue would be useful to the CTRW community.

**RESPONSE:** We do not claim that the v_T parameter in the CTRW should correspond to the value of v_W. In fact, we do not invoke v_W in the CTRW, and thus there is no "mismatch" or inconsistency between these two parameters in the CTRW framework. (These aspects are described in detail in CTRW literature cited in the manuscript; it seems inappropriate to review CTRW in great detail in the current manuscript.) Rather, we emphasize that the assessment and use of v_T and v_W is a matter of general interest, and of relevance to virtually any modelling approach.

In particular, we emphasize that the fundamental formulation of the ADE *requires* that the velocity term in the equation correspond to – i.e., be identical to – v_W. Every textbook development of the ADE immediately invokes the mean linear water velocity, v_W, based on Darcy's law. Thus, use of v_T, as estimated from a breakthrough curve, for example, as the value of "*v*" in the ADE is fundamentally incorrect. Otherwise, one is using a "circular argument", inserting a mean tracer velocity, which already takes into account the influences of dispersion and diffusion, into the ADE, and then attempting to fit a full breakthrough curve by solution of the ADE with an additional fitting parameter (dispersivity or dispersion coefficients). Even with this approach, we show in the manuscript that such a fitting approach with the ADE cannot match the measurements, particularly the long tailing behavior. It is at this point that we consider a CTRW anomalous transport interpretation of the system dynamics to successfully fit and interpret the measurements.

**_DONE:_** In light of the referee comments and our responses above, we have added a statement in the revised manuscript to clarify the arguments already provided therein. We added (Section 3.3., first paragraph): "It should be emphasized that the distinction between v_W and v_T holds regardless of the choice of model applied to interpret breakthrough curves. For example, the derivation of the classical ADE, and variants thereof, in particular, is predicated on v_W. In contrast, the continuous time random walk framework (CTRW) formulation discussed below is essentially founded on v_T."

Meanwhile, the community that uses isotopes to study catchment properties is moving on from the notion of 'mean travel time'. The leading-edge approaches do not require it, and it is reported less often in favor of other more reliable metrics, like those based on storage selection.

**RESPONSE:** We acknowledge that at least some portion of the community is now working more extensively with storage selection theory. However, another portion of the community continues to work with mean travel times in spatially 1D domains, as seen in the literature. We therefore believe it remains valuable and important to clearly report the discrepancies and inconsistencies that we address in the manuscript, and to illustrate the impact on assessment of aquifer storage thickness, which remains a key issue in catchment hydrology studies.

I would note in passing that the authors do not seem familiar with the storage selection approach. They seem to be under the impression that it is based on the collapse of the system to one spatial dimension (Line 331). This is not the case -- in storage selection theory the system is collapsed to zero spatial dimensions.

**RESPONSE:** Reference to 1D interpretations of catchment and aquifer conceptualizations was not intended to refer specifically to storage selection theory. In fact, we did not explicitly refer to the storage selection approach at any point in the manuscript; we certainly did not claim, nor intend to claim, that the storage selection approach collapses to one spatial dimension (according to the referee's reference to line 331). This latter approach is different, and we in fact do not refer to it explicitly in the manuscript.

*DONE:* In light of the above two referee comments, and our responses, we have added a note in the paragraph containing the reference to consideration of a system under one spatial dimension (Line 331 in the original manuscript). To add perspective, we now include the text: "An alternative approach employs storage selection theory, which involves collapsing the system to zero spatial dimensions and defining functions that interpret age-ranked release of water from storage and exit from the catchment, or in other words, defining functions that quantify the probability of water of a certain age being discharged at a given time."

---

## Author Comment (AC3)

**Response to referee #2 – second response:**

I strongly urge the authors to reconsider their statement: "We thus show that v_T, the mean travel time of the isotope and the tracer, is distinct from the mean travel time of the "water" itself, as determined from Darcy's law." On first reading this appears to be saying "the mean travel time of the water is distinct from the mean travel time of the water". I understand this isn't their intended meaning, but this framing is at best obscure and might in fact be read as deliberately provocative.

**RESPONSE:** The quoted statement is from our Reply to the referee's first comment. The specific wording in the original manuscript, and further reinforced in the revised manuscript, is slightly different. In the manuscript itself, we state: "By estimating the porosity and cross-sectional area of flow through the column, and for a fixed $Q$, $\bar{v}_w$ can be calculated using Darcy's law, and the apparent [*this word added in the revised version*] mean travel time of water through the column can be estimated by dividing the column length by $\bar{v}_w$." The discussion regarding v_T and v_W first appears in Section 3.2, and the definitions of these two terms are stated clearly and simply. The mean velocity of a tracer, or isotope, can indeed be different from the apparent mean water velocity. There is nothing obscure or intentionally provocative in this – rather, this is a fundamental point that is unfortunately not always recognized when interpreting and quantifying dynamics of fluid flow and chemical transport.
***DONE*:** In the revised manuscript, we added the word "apparent", as noted above, for clarity.

First of all, Darcy's Law (as I have always understood it) is a statement about the relationship between a pressure gradient and bulk water flux (volume per area per time). The present paper never reports, calculates, or relies on a pressure gradient, and so they don't appear to be making use of Darcy's Law in a way that I can recognize. Consider: the quantity v_W obtained from v_W=Q/(nA) can be calculated regardless of whether the flow in the porous media is laminar (and so Darcy's law would be expected to hold) or turbulent (in which case it would not). v_W is therefore quite independent of Darcy's Law.

**RESPONSE:** As we discuss in the manuscript (Section 2, Methods), we in fact prescribed the volumetric flow rate, $Q$, in the experiments, and state the values. The volumetric flow rate relies

directly on the pressure gradient, so that the expression v_W=Q/(nA) is indeed Darcy's law and applicable to the experiments we report. Yes, we agree that one can in principle (mis)apply Darcy's law to situations of turbulent flow, but this is not the case in our experiments, and in most others reported in the literature. In other words, the v_W that we calculate, and that is usually reported in other studies in the literature, is indeed representative of and based on Darcy's law; the v_W we report is definitely not "quite independent of Darcy's Law".

***DONE***: In the revised manuscript, we added (line 134) the definition of $Q$ as the volumetric flow rate (which appeared later), and added "$Q$" to the heading in Table 1, for added clarity.

Second, the authors seem to want to have their cake and eat it too, when it comes to the relationship between v_T and v_W. Consider these two statements:

"In particular, we emphasize that the fundamental formulation of the ADE *requires* that the velocity term in the equation correspond to – i.e., be identical to – v_W. Every textbook development of the ADE immediately invokes the mean linear water velocity, v_W, based on Darcy's law."

-- fine, so when the ADE applies we would expect v_W=v_T. Deviations from that would indeed be surprising, but that is not what was observed here (since the ADE does not apply to the data presented).

**RESPONSE:** We agree and indeed suggest in the manuscript, similar to the referee, that the ADE does apply when v_W=v_T. However, it should be noted that literature over the last 25+ years shows numerous examples of experiments (and numerical simulations) in a wide range of porous (and fractured) media, over a wide range of spatial and temporal scales, that exhibit non-Fickian (non-ADE) behavior --- in other words, deviations between v_W and v_T are actually very common. The difficulty is that the ADE is frequently assumed to hold, when in fact it does not for the situation and measurements under analysis.

"In fact, we do not invoke v_W in the CTRW, and thus there is no "mismatch" or inconsistency between these two parameters in the CTRW framework" and "the continuous time random walk framework (CTRW) formulation discussed below is essentially founded on v_T"
-- fine, so when the ADE does not apply and we have to use CTRW, we would not expect v_W=v_T, as their meaning diverges. v_T is a parameter of the CTRW conceptual framework,

and within that framework it is conceptually distinct from v_W. They coincide only when the CTRW reduces to the ADE.

In the present dataset the ADE does not apply ("the ADE cannot match the measurements, particularly the long tailing behavior") and so (by the author's logic) we should not expect v_W=v_T, and **indeed this is the case**.

So where is the mystery here?

**RESPONSE:** The referee again agrees with us, but then concludes that there is no mystery here. In response, we emphasize that it is misleading to state that "v_T is a parameter of the CTRW conceptual framework" – it is a general concept (see Sect. 3.2, which describes a Gedanken experiment) that is relevant in all transport studies and modelling efforts. Moreover, further to the Response above, if ADE behavior is actually not exhibited in such a wide range of porous (and fractured), then why is the ADE almost "automatically" assumed to be the correct assessment of the transport behavior and then applied? As discussed in the manuscript, application of the ADE in aquifer and catchment studies remains ubiquitous. And as discussed in the manuscript, we demonstrate how this can lead to serious over-estimation of aquifer storage thickness.

It seems like the primary issue point being made in this paper is about how in porous media sufficiently heterogeneous as to be non-fickian the quantity v_W=Q/(nA) cannot be naively interpreted as the 'mean velocity' of the water. Instead, the presence of long tails adds some important nuance and complexity to the very notion of "mean velocity".

That might be a useful point to make, but the weird distinction the authors draw between the "velocity of the water" and the "velocity of the water isotopes" rather obscures it. Also, I would note again that the phenomenon at issue appears to be the case *regardless of what tracer is used*, so I'm still not sure why isotopes are being singled out.

**RESPONSE:** As the referee states, we agree that the presence of long tails adds important complexity to the very notion of mean velocity. This important complexity and the fact that it needs to be recognized, both conceptionally and for the sake of different estimates of aquifer properties – e.g., thickness – is indeed a main motivation for this manuscript. However, we respectfully disagree with the statement: "the weird distinction the authors draw between the "velocity of the water" and the "velocity of the water isotopes"". There is nothing weird, except

for the realization that this distinction is often unrecognized, so that modelling and interpretation efforts are biased and incorrect.

Our point, as demonstrated and discussed theoretically, experimentally, and using model simulations, is that there is a distinction between the apparent mean water velocity and the velocity of any measurable tracer moving with the water. While this point would likely be accepted by most researchers when discussing chemical tracers, we argue that there is a misconception when comparing the apparent mean water velocity and the mean velocity of water isotopes. Water isotopes are commonly thought to *exactly represent* the apparent mean water velocity; but "tagging" of individual water molecules identifies them similarly to chemical tracers, so that the misconception noted here is often overlooked. This misconception then propagates to incorrect use of the ADE and estimates of catchment properties such as aquifer storage thickness.
* * *
**_PLEASE NOTE THAT VARIOUS REVISIONS TO THE MANUSCRIPT ARE NOTED IN EACH OF THE TWO RESPONSES TO REFEREE #2._**

---

## Author Response (AR2)

First, we note that a fourth referee (first round of reviews) provided a highly positive appraisal of the manuscript and recommended only minor revisions which we addressed previously. No further review comments were uploaded by this referee.

**Response to Referee #1, Report #3:**

Nothing to suggest. Please see the recommendation to the editor

**RESPONSE:** We thank the referee for the positive recommendation for publication.

**Response to Referee #2, Report #1:**

The central claim of this paper, as stated in the abstract, is that "For both water isotopes and inert chemical tracers, the measured mean tracer velocity is not necessarily equal to the apparent mean water velocity". However it does not make sense to claim both of the following statements are true simultaneously:

- 1. The mean velocity of the tracer v T is different from the mean velocity of the water v w
- 2. The tracer is water molecules or moves identically to water molecules

The only way I can make sense of this paper is if v\_T is not the mean tracer velocity, but is rather a scaling parameter of the CTRW model that has dimensions of velocity (but is not the mean). However during our review discussion the authors rejected this suggestion, and claim that v\_T is in fact the mean tracer velocity. Therefore I cannot see how this paper is conceptually coherent.

**RESPONSE:** The referee appears to have accepted our responses and clarifications to all but one of the comments raised in the two previous replies. Here, the referee first notes a central clear claim ("...the measured mean tracer velocity is not necessarily equal to the *apparent* mean water velocity") and then lists two points that are claimed to be contradictory. But in listing these two points, the referee ignores the preceding sentence with the "central claim". We do *not* claim statement #2 above – rather, we claim that the average velocity of the *tagged* water molecules (i.e., the water isotopes) is not identical to the *apparent* mean water velocity.

At this point, the referee focuses on movement of water molecules, whether pure  $H_2O$  or deuterium isotopes. The referee misses the key recognition, as we state clearly in the manuscript, that the very act of tagging water molecules yields a measurement representative of their migration as a chemical tracer. This is stated clearly, for example, in

(i) Section 3.2, below Figure 2 describing the Gedanken experiment: "The act of tagging water effectively changes a water molecule to act as a "non-water" tracer, in the context of breakthrough measurements; in other words, the measured velocity represents the mean velocity of a tracer (be it a water isotope or an inert chemical), rather than the apparent mean velocity of the water."

and

(ii) Conclusions: "Notably, water isotopes exhibit the same transport behavior as tracers; the very act of tagging water molecules, implicit in the measurement of any water isotope, yields a measurement of their migration as a chemical tracer, which is not identical to the bulk water flow."

**DONE:** In the revised manuscript, we have also noted this point in the Abstract, to add further clarity.

**Response to Referee #3, Report #2:**

**Overall Evaluation**

This manuscript presents a carefully designed laboratory study that addresses a fundamental and widely held assumption in catchment hydrology: that water isotopes (e.g.,  $D_2O$ ) behave identically to the bulk water molecules ( $H_2O$ ) and can thus be used directly to infer the mean water velocity and transit times. The experimental work is robust, comparing breakthrough curves of deuterium-enriched water and bromide tracer under various flow conditions and porous media configurations. The key finding—that water isotopes exhibit transport behavior indistinguishable from that of an inert chemical tracer, leading to a mean tracer velocity ( $v\bar{v}$ ) that is measurably slower than the apparent mean water velocity ( $v\bar{v}$ )—is significant and has direct implications for the interpretation of field-scale isotope data.

The use of the CTRW framework to quantify the non-Fickian transport and the clear demonstration of the inequality  $v\bar{v}w \neq v\bar{v}T$  are particular strengths. The discussion linking these laboratory findings to potential overestimations of aquifer storage thickness in catchment studies is timely and relevant. The manuscript is generally well-written and structured. However, to strengthen the impact and clarity of the work, several aspects require further elaboration and clarification.

**RESPONSE:** We thank the referee for the positive evaluation and recommendation for publication. We appreciate the highly constructive comments, which we address below point-by-point. We are pleased to incorporate additional explanations for all comments in the revised manuscript.

**Specific Comments**

(1) The manuscript rightly concludes that "the very act of tagging water molecules... yields a measurement of their migration as a chemical tracer." This is a crucial point. It should be explicitly discussed whether and how the rapid equilibrium isotope exchange between H2O, HDO, and D2O (mentioned in the methods) influences this "tagging" concept. Does this equilibrium exchange mean that the "tag" is effectively transferred between molecules, potentially altering the perceived transport behavior? A brief discussion on how this isotopic exchange is accounted for in the interpretation of the D2O breakthrough curves would strengthen the argument that it behaves as a conservative solute and not as a perfect proxy for the bulk water movement.

**RESPONSE:** We explain in the Methods section, last paragraph, that "In excess of  $H_2O$ ,  $D_2O$  rapidly converts to HDO in equilibrium ( $D_2O + H_2O \rightleftharpoons 2HDO$ ), which is linearly correlated to the measurable  $ArD^+$  ion in the plasma." We note that analytical method measures elemental (not molecular) concentrations, so that the signal mirrors the *total* amount of the analyte(s), in this case the total  $ArD^+$  in the sample. We also then note that, as a result, the equilibrium exchange of deuterium in the water is accounted for implicitly, with the BTCs shown in Fig. 1 representing effective measurements and the overall transport behavior.

**DONE:** In the revised manuscript, we have (i) added an explanation that we measure the total amount of deuterium in the samples (lines 163-165, revised manuscript), and (ii) in Section 3.1, end of the first paragraph (lines 189-194, revised manuscript), we expand the discussion to note that how this isotopic exchange is accounted for in the interpretation of the  $D_2O$  breakthrough curves.

(2) The study uses two packing configurations (homogeneous and heterogeneous) and two flow rates. While this is a good start, a more detailed justification for the specific grain sizes, the nature of the heterogeneity created, and the selected flow rates would be beneficial. For instance, what specific pore-scale structures or inclusion characteristics were targeted to induce the observed anomalous transport?

**RESPONSE:** We explained in Section 2, paragraph 2, our reasons for choosing the configurations and sand types. The setups are based on previous results from Elhanati et al. (2023); investigating similar types of experimental setups offers continuity and prior support that the transport is inherently non-Fickian. It is important to recognize, as shown repeatedly in the literature, that even pore-scale heterogeneity in macroscopically uniform sand columns, over a range of flow rates and travel distances, can induce non-Fickian transport. **DONE:** In the revised manuscript, we have added this information at the end of Section 2, paragraph 2, lines 124-127.

(3) The manuscript states that the breakthrough curves for Br- and D2O are "similar." This similarity is visually apparent in the figures but could be further substantiated quantitatively. It is recommended to include a quantitative metric, such as the calculation of the temporal moments (e.g., mean arrival time, variance) or a statistical goodness-of-fit measure between the Br- and D2O curves for key experiments. This would provide a more objective and robust basis for the central claim of identical transport behavior.

**RESPONSE:** We calculated (but had not provided the metrics) previously, to support our claims regarding similarity of the BTCs shown in Figure 1. Thank you for suggesting this. **DONE:** In the revised manuscript, we now include Table 2, following Figure 1, which provides the suggested metrics (mean travel times, standard deviations and correlation coefficients to compare each pair of BTCs), and reference to the new Table on line 197.

(4) The discussion in Section 3.4 is insightful but could be expanded. The extrapolation from a 1D laboratory column to a 3D catchment is a significant step. The authors should more explicitly discuss the potential limitations and necessary conditions for this scaling. For example, how might the relative impact of  $\bar{v}w$  vs.  $\bar{v}T$  differ in a 3D system with more complex flow paths, recharge dynamics, and the presence of unsaturated zones?

Acknowledging these complexities would provide a more nuanced perspective on the generalizability of the laboratory findings.

**RESPONSE:** We agree that expanding from a 1D laboratory setup to a 3D catchment is significant. We therefore tried to be "modest" in the analysis and in clearly stating limitations in the analysis, as they appear in the first paragraph of Section 3.4. Clearly, the presence of highly complex flow paths, recharge dynamics, and partially water-saturated regions in catchments will impact the relative difference between  $\bar{v}_w$  and  $\bar{v}_T$ . This aspect remains to be investigated in detail, but it can be speculated that the added complexity might lead to an even wider range of sources for tracer retention, as suggested by some field-scale analyses of anomalous transport (e.g., Goeppert et al., 2020; Dentz et al., 2023).

**DONE:** In the revised manuscript, we have added these comments in Section 3.4, end of paragraph 2, lines 397-402.

(5) The manuscript notes the slightly different molecular weights (and hence diffusion coefficients) of H2O and D2O but dismisses this as a significant factor for the short-term experiments. Given that the study fundamentally challenges the assumption of identical behavior, a more thorough discussion of this point is warranted.

**RESPONSE:** A brief reference to molecular weights appears in Section 3.4, paragraph 5, Lines 418-421. In light of the referee's comment, we note that over relatively long time scales, the impact of different rates of diffusion may also be significant: diffusion on one hand can enhance tracer trapping and thus extend retention times in low-permeability zones, while on the other hand lead to increased uniformity of the tracer plume concentrations. We note that the specific impact of molecular diffusion in these scenarios remains to be investigated.

**DONE:** In the revised manuscript, we have added these comments in Section 3.4, end of paragraph 2, lines 421-426.